

# Ellipse packing in two-dimensional cell tessellation: a theoretical explanation for Lewis's law and Aboav-Weaire's law

Kai Xu

Fisheries College, Jimei University, Xiamen, China

## ABSTRACT

**Background**. Lewis's law and Aboav-Weaire's law are two fundamental laws used to describe the topology of two-dimensional (2D) structures; however, their theoretical bases remain unclear.

**Methods**. We used R software with the Conicfit package to fit ellipses based on the geometric parameters of polygonal cells of ten different kinds of natural and artificial 2D structures.

**Results**. Our results indicated that the cells could be classified as an ellipse's inscribed polygon (EIP) and that they tended to form the ellipse's maximal inscribed polygon (EMIP). This phenomenon was named as ellipse packing. On the basis of the number of cell edges, cell area, and semi-axes of fitted ellipses, we derived and verified new relations of Lewis's law and Aboav-Weaire's law.

**Conclusions**. Ellipse packing is a short-range order that places restrictions on the cell topology and growth pattern. Lewis's law and Aboav-Weaire's law mainly reflect the effect of deformation from circle to ellipse on cell area and the edge number of neighboring cells, respectively. The results of this study could be used to simulate the dynamics of cell topology during growth.

## INTRODUCTION

A two-dimensional (2D) plane can be tessellated by convex polygons. Scientists are interested in natural and artificial 2D structures that share the common feature that the coordination number of vertices (the number of edges meeting at a vertex) of polygonal cells always equals three. The cell topology of these 2D structures can be described according to three laws: Euler's law, Lewis's law, and Aboav-Weaire's law (*Weaire & Rivier, 1984*). The latter two laws were first observed empirically by Lewis and Aboav, with the original aims of understanding laws in biological and physical structures, respectively (*Aboav, 1970*; *Lewis, 1926*; *Lewis, 1928*; *Weaire, 1974*). Although Lewis's law and Aboav-Weaire's law are essential for understanding the formation mechanisms of 2D structures, their theoretical explanations are deficient (*Mason, Ehrenborg & Lazar, 2012*; *Weaire & Rivier, 1984*). The coordination number is a short-range order that mathematically determined that the average number of edges per cell is six (*Graustein, 1931*).

Corresponding author
Kai Xu, kaixu@jmu.edu.cn

When this study restricts attention to biological 2D structures, the word ''cell'' represents the top and bottom faces of a prismatic cell. The dynamics of cell topology during growth make biological 2D structures even more complicated than other types of 2D structures. For example, internal angles of *Pyropia haitanensis* cells have been concentrated in the range of 100–140° by direction-specific division and direction turning of cell edges, which suggested that the cells tended to form regular polygons (*Xu et al., 2017*). These observations hinted at the possibility of undiscovered short-range orders in 2D structures. A recent study by *Xu, Hutchins & Gao (2018)* found that the effective coverage area of ellipse-shaped exoskeletons of microalga *Emiliania huxleyi* cells tended to approach the maximal area of an ellipse's inscribed polygon (EIP). This study identified a similar phenomenon: the polygonal cells of natural and artificial 2D structures were inclined to form the ellipse's maximal inscribed polygon (EMIP). On the basis of this short-range order, the present study derived and verified new relations of Lewis's law and Aboav-Weaire's law.

## MATERIAL AND METHODS

We used Amscope Toupview 3.0 software to analyze the images of ten kinds of 2D structures. The images of *P. haitanensis* and two images of onion were taken by the author of this study, and the others were derived from the published papers. The nonliving biological 2D structures included the following: cross-sections of shells of *Atrina rigida*, *Atrina vexillum*, and *Pinna nobilis* (*Reich et al., 2018*). The living biological 2D structures included the following: epidermal tissues of *Agave attenuate*, *Allium cepa* (onion), and *Allium sativum* (garlic) (*Mombach, Vasconcellos & De Almeida, 1990*); and *P. haitanensis* thalli. The physical 2D structures include amorphous silicon dioxide ($SiO_2$) film (*Büchner et al., 2016*) and soap (*Aboav, 1980*). The random-seeded Voronoi diagrams are artificial 2D structures that also have been used for analysis (*Aboav, 1985*). For each polygonal cell, we measured the area ($A_C$), coordinates of center ($X_{PC}$, $Y_{PC}$), and vertices ($X_V$, $Y_V$). We used R software (*R Core Team, 2018*) with the Conicfit package to fit an ellipse based on the coordinates of the vertices of each polygonal cell (Fig. 1A) (*Chernov, Huang & Ma, 2014*). For the $SiO_2$ film, the vertices of polygonal cells were formed by the silicon (Si) atoms. Five geometric parameters could be used to describe the ellipse, which include the semi-major axis *a*, semi-minor axis *b*, coordinates of center ($X_{EC}$, $Y_{EC}$), and angle of tilt of the major axis $\theta$ (Fig. 1B). On 2D geometry, five points determine a conic; for example, the ellipse. For polygons with five or more edges, we set $X_{PC}$ and $Y_{PC}$ as the initial values of the coordinates of the ellipse center to improve fitting. As for cells with only four edges, we combined the coordinates of the four vertices and the four midpoints of the edges as a single data set to fit an ellipse in the same manner as the cells with five or more edges. Then, we set the geometric parameters of the fitted ellipse as the initial values to fit the second ellipse for the coordinates of the four vertices. We found the second ellipse to be the smallest one among all of the fitted ellipses, and which we used for analysis. We provide our reasons for finding the smallest circumscribed ellipses for four-edged polygonal cells in the next section.
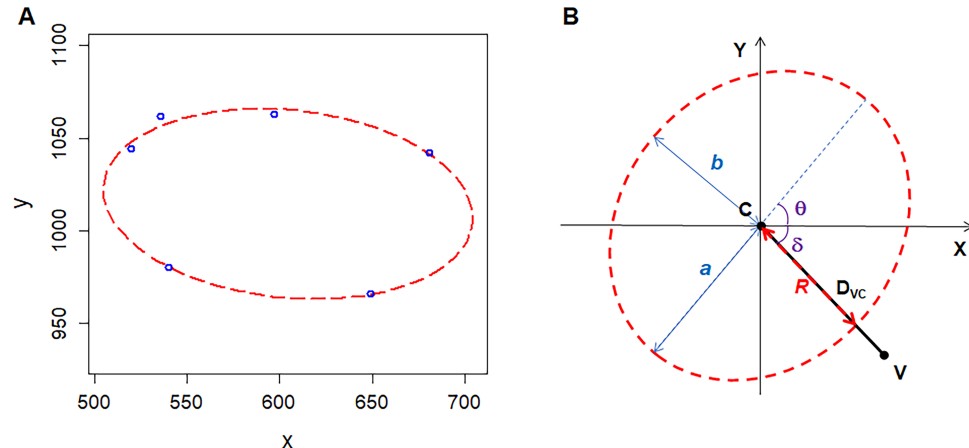

**Figure 1** **Geometry of polygonal cell and fitted ellipse.** (A) Coordinates of the vertices of a polygonal cell and fitted ellipse. We plotted the ellipse using R software plus the Conics package (*Chernov, Huang & Ma, 2014*). (B) A diagram shows semi-major-axis ($a$), semi-minor-axis ($b$), angle ($\delta$) between line VC and $X$-axis, angle ($\theta$) of tilt of the major axis, distance ($D_{VC}$) between the center of the ellipse and a vertex of polygonal cell, distance ($R$) from the center of the ellipse to the cross point of line VC and fitted ellipse .

We calculated the area of the ellipse ($A_E$) as follows:

$$A_E = \pi ab, \tag{1}$$

The area of the maximal inscribed polygon of the ellipse ($A_{MIP}$) is

$$A_{MIP} = 0.5nab\sin\left(\frac{2\pi}{n}\right), \tag{2}$$

where $n$ is the number of edges of inscribed polygon (*Su, 1987*). The form deviation of vertex (*FD*) is

$$FD = \frac{D_{VC} - R}{R} \times 100\%, \tag{3}$$

where $D_{VC}$ is the distance between a vertex and the center of the fitted ellipse (the length of line VC)

$$D_{VC} = \sqrt{(X_V - X_{EC})^2 + (Y_V - Y_{EC})^2}, \tag{4}$$

$R$ is the distance from the ellipse center to the cross point of the fitted ellipse and the line VC

$$R = \frac{ab}{\sqrt{(a\sin(\arctan(\tan\theta) - \delta))^2 + (b\cos(\arctan(\tan\theta) - \delta))^2}}, \tag{5}$$

where $\delta$ is the angle between line VC and $X$-axis, the ranges of $\theta$ and $\delta$ are $[0, \pi)$ and $(-0.5\pi, 0.5\pi)$, respectively (Fig. 1B). R code (Document S1) and three examples (Data S1) for these calculations are included in the supplementary files.

**Table 1  Parameters of polygonal cells of *P. haitanensis* and fitted ellipses.**

| Parameters | Mean ± SD | Range | Sample number |
|---|---|---|---|
| Average number of cell edges | $6.03 \pm 0.88$ | 4–10 | 1375 |
| Form deviation ($FD$, %) | $0.00 \pm 3.14$ | $-13.94$–20.56 | 8291 |
| Fitted semi-major-axis ($a$, $\mu$m) | $19.86 \pm 2.76$ | 13.91–37.22 | 1375 |
| Fitted semi-minor-axis ($b$, $\mu$m) | $15.34 \pm 1.72$ | 9.29–21.76 | 1375 |
| $a/b$ | $1.31 \pm 0.21$ | 1.01–2.71 | 1375 |
| Area of fitted ellipse ($A_E$, $\mu$m$^2$) | $961.80 \pm 195.25$ | 506.16–2016.86 | 1375 |
| Area of the maximal inscribed polygon of fitted ellipse ($A_{MIP}$, $\mu$m$^2$) | $788.19 \pm 172.00$ | 343.24–1667.93 | 1375 |
| Area of cell ($A_C$, $\mu$m$^2$) | $705.98 \pm 148.98$ | 303.94–1512.63 | 1375 |
| $A_C/A_{MIP}$ | $0.90 \pm 0.07$ | 0.48–1.00 | 1375 |

## RESULTS AND DISCUSSION

### Ellipse packing

Thallus of red alga *P. haitanensis* is a single-layered prismatic cell sheet that is a mathematical consequence of 2D expansion on a plane by cell proliferation (*Xu et al., 2017*). Thus, *P. haitanensis* thalli can be simplified as 2D structures. We found the average number of edges of *P. haitanensis* cells to be $6.0 \pm 0.9$ (1375 cells in 13 thalli were examined; Table 1), which was consistent with previous studies on *P. haitanensis* as well as studies on many other organisms and physical structures (*Gibson et al., 2006*; *Sánchez-Gutiérrez et al., 2016*; *Weaire & Rivier, 1984*; *Xu et al., 2017*). According to Euler's 2D formula, this kind of phenomenon has been mathematically determined when the coordination number of each vertex equals three when different-sized cells tessellate a 2D plane (*Graustein, 1931*; *Weaire & Rivier, 1984*). The size differences between cells indicated that these 2D structures display a long-range disorder, because the unit cell has neither periodicity nor translational symmetry. In addition, the average number of edges of *P. haitanensis* cells quickly approached six with an exponential increase in cell number resulting from an increase in body size (*Xu et al., 2017*). Thus, this phenomenon has been observed only when the 2D structures contain a large number of cells (*Graustein, 1931*; *Lewis, 1926*; *Weaire & Rivier, 1984*).

This study found that the vertices of cells of *P. haitanensis* could be used to fit ellipses with an average form deviation of $0.00 \pm 3.14\%$ (8,291 vertices in 1375 cells were examined; Table 1; Data S2). We found similar results in the other 2D structures (Table 2; Data S3). Thus, the polygonal cells of 2D structures could be considered to be EIPs, which ensured that all of the cells were convex polygons. The ratios of the area of the cell and fitted EMIP ($A_C/A_{MIP}$) of *P. haitanensis* ranged from 0.48 to 1.00 with an average value of $0.90 \pm 0.07$ (Table 1), and 90% of the values were concentrated in a range from 0.78 to 0.97 (Data S2). The random-seeded Voronoi diagrams and three kinds of epidermal tissues showed a similar average ratio of $A_C/A_{MIP}$. The $A_C$ of amorphous $SiO_2$, cross-sections of mollusk shells, and soap, however, were very close to $A_{MIP}$ (Table 2; Data S3). Thus, we divided the 2D structures into three categories based on $A_C/A_{MIP}$: Type I, monohedral tiling using

**Table 2** **Parameters of polygonal cells of 2D structures.** Parameters $a/b$, $A_C/A_{MIP}$, and $FD$ represent the ratio of fitted semi-major-axis/semi-minor- axis, ratio of cell area/EMIP, and form deviation, respectively. Except for the last 2D structure, we derived the images of the others from published papers: amorphous $SiO_2$ (*Büchner et al., 2016*), cross-sections of mollusk shells (*Reich et al., 2018*), soap (*Aboav, 1980*), Voronoi diagrams (*Aboav, 1985*), epidermal tissues of *Agave attenuate*, *Allium cepa* (onion), and *Allium sativum* (garlic) (*Mombach, Vasconcellos & De Almeida, 1990*). Sample numbers are shown in parentheses.

|  | **2D structures** | $a/b$ | $A_C/A_{MIP}$ | $FD$ (%) |
|---|---|---|---|---|
| Type II | Amorphous $SiO_2$ | $1.20 \pm 0.12$ (10) | $0.99 \pm 0.01$ (10) | $0.00 \pm 3.73$ (62) |
|  | Cross-sections of mollusk shells | $1.14 \pm 0.07$ (30) | $0.97 \pm 0.02$ (30) | $0.00 \pm 0.89$ (170) |
|  | Soap | $1.11 \pm 0.05$ (20) | $0.98 \pm 0.01$ (20) | $0.00 \pm 1.14$ (118) |
| Type III | Voronoi diagrams | $1.43 \pm 0.25$ (50) | $0.87 \pm 0.09$ (50) | $0.01 \pm 2.48$ (286) |
|  | *Allium sativum* (garlic) | $3.24 \pm 0.78$ (10) | $0.95 \pm 0.03$ (10) | $0.01 \pm 2.46$ (56) |
|  | *Allium cepa* (onion) | $3.43 \pm 1.03$ (10) | $0.92 \pm 0.03$ (10) | $0.03 \pm 4.58$ (57) |
|  | *Agave attenuat* | $1.13 \pm 0.06$ (10) | $0.98 \pm 0.01$ (10) | $0.00 \pm 1.14$ (60) |
|  | *Allium cepa* (onion) | $1.94 \pm 0.38$ (20) | $0.95 \pm 0.04$ (20) | $0.00 \pm 1.84$ (113) |

six-edged EMIPs (e.g., tile by regular hexagons); Type II, tiling using different-sized and different-edged EMIPs (e.g., 2D amorphous $SiO_2$); and Type III, tiling using different-sized and different-edged EIPs. For amorphous $SiO_2$, the bond length should be different to obey the ellipse packing. For the Types I and II 2D structures, $A_C$ equals $A_{MIP}$.

These results suggested that the fitted ellipse should be the smallest circumscribed ellipse of the polygonal cell, which was the reason we sought to find the smallest ellipse for four-edged cells in this study. A recent study reported similar phenomenon on single-celled microalga *E. huxleyi* (*Xu, Hutchins & Gao, 2018*). *E. huxleyi* cells were fully covered by interlocking calcite exoskeletons, and the specific geometry of exoskeletons resulted in the effective coverage area of exoskeletons tending to reach the maximal area of an inscribed polygon of ellipse-shaped exoskeletons.

Obviously, the effects of growth on cell topology for these three types of 2D structures were quite different. For 2D structures made of EMIPs, if the variations in topology were achieved by reconstruction or by transition to other types, the topological variations of all of the cells had to be finished synchronously to obey ellipse packing. Otherwise, we observed a cell area less than $A_{MIP}$. For example, the areas of polygonal cells were always equal to $A_{MIP}$ during the evolvement of soap (Table 2) (*Aboav, 1980*). As for 2D structures made of different-sized and different-edged EIPs, most of the cells were smaller than their corresponding EMIPs. For biological 2D structures, complicated life activities strongly altered the cell size and topology (e.g., accumulation of organic components, respiration, cell division and fusion, water metabolism, and exposure of stressful conditions). Moreover, different metabolism rates between cells and the asynchronous cell cycle would make the dynamic behaviors of cell topology even more complicated. Based on geometric limits, *Xu, Hutchins & Gao (2018)* proposed that the regular dodecahedron-shaped cells of coccolithophore *Braarudosphaera* spp. should be the resting or cyst stage of the life cycle. Similarly, the cells of living biological 2D structures should be EIPs rather than EMIPs, which suggested that living biological 2D structures belonged to Type III (Tables 1–2) and that specific cases may have manifested such that the complicated life activities would not

influence the cell topology like the regular polyhedral cells of *Braarudosphaera* spp. The variations in the cell topology of Type III 2D structures had to be achieved by fine-tuning. In the following sections, we detail the effects of growth on cell topology for all three types of 2D structures.

The eccentric angle of neighboring vertices of an *n*-edged EMIP is equal to $2\pi/n$ (*Su, 1987*). Therefore, the eccentric angles of a six-edged EMIP equal to 60° and the average internal angle is 120°. On the basis of observations of direction-specific divisions (which resulted in equal-sized divisions) and division-associated direction changes of the cell edges (concentrated internal angles ranging from 100° to 140°), *Xu et al. (2017)* found that *P. haitanensis* cells tended to form regular polygons. The closer the polygonal cell was to a regular polygon, the closer the cell was to a spherical shape, which could help maintain force balance (*Chen, 2008*; *Ingber, Wang & Stamenovic, 2014*). Unbalanced forces could result in unequal-sized cell division (*Kiyomitsu, 2015*). Equal-sized daughter cells, however, were always found in the cell proliferation of *P. haitanensis* thalli (*Xu et al., 2017*).

## Lewis's law

Lewis's law is an empirical law that suggests that $A_C$ of an *n*-edged cell is related linearly to *n* (*Chiu, 1995*; *Lewis, 1926*; *Lewis, 1928*; *Weaire & Rivier, 1984*). According to Eq. (2), the cell area of Type II 2D structures increased with edge number. To investigate the relationship between the number of edges and the cell area of Type III 2D structures, we used *P. haitanensis* thalli as the research material. The average values of $A_E$, $A_{MIP}$, and $A_C$ increased with *n*, whereas the difference between the average values of $A_E$ and $A_C$ decreased (Fig. 2A). Except for $n > 8$, the average ratios of *a/b* were stable regardless of the values of *n* (Fig. 2B). Because $A_{MIP}$ is $\frac{n}{2\pi}\sin\left(\frac{2\pi}{n}\right)$ times $A_E$ (*Su, 1987*), the ratio of $A_{MIP}/A_E$ approaches one with an increase of *n* (Fig. 2C). We found positive linear relationships between $A_C$ and $A_E$ ($R^2 = 0.73$, $P < 0.0001$; Fig. 2D) and between $A_C$ and $A_{MIP}$ ($R^2 = 0.85$, $P < 0.0001$, Fig. 2E). Thus, $A_C$ can be calculated by the following empirical equation:

$$A_C = 0.80A_{MIP} + 78.79 = 0.40nab\sin\left(\frac{2\pi}{n}\right) + 78.79, \tag{6}$$

where the maximal value of $n\sin\left(\frac{2\pi}{n}\right)$ is

$$\lim_{n\to\infty} n\sin\left(\frac{2\pi}{n}\right) = 2\pi. \tag{7}$$

Because both $n\sin\left(\frac{2\pi}{n}\right)$ and $A_E$ increase with *n* (Figs. 2A and 2B), $A_C$ also increased with *n*, which was consistent with Lewis's law. Overall, the present study suggested that the relationship between $A_C$ and *n* is more complex than previous believed.

The cell proliferation of biological 2D structures can be used as a window to observe the dynamic behavior of cell topology during growth. By equal-sized division, mitosis shall strongly disturb cell topology. Obviously, division should separate a cell along the direction of the minor-axis of the fitted ellipse, making daughter cells closer to EMIPs (Fig. 3A). Nearly 150 years ago, Hofmeister proposed a similar idea called long-axis division (*Hofmeister, 1863*). More complicated, however, *Xu et al. (2017)* found that divisions preferred to
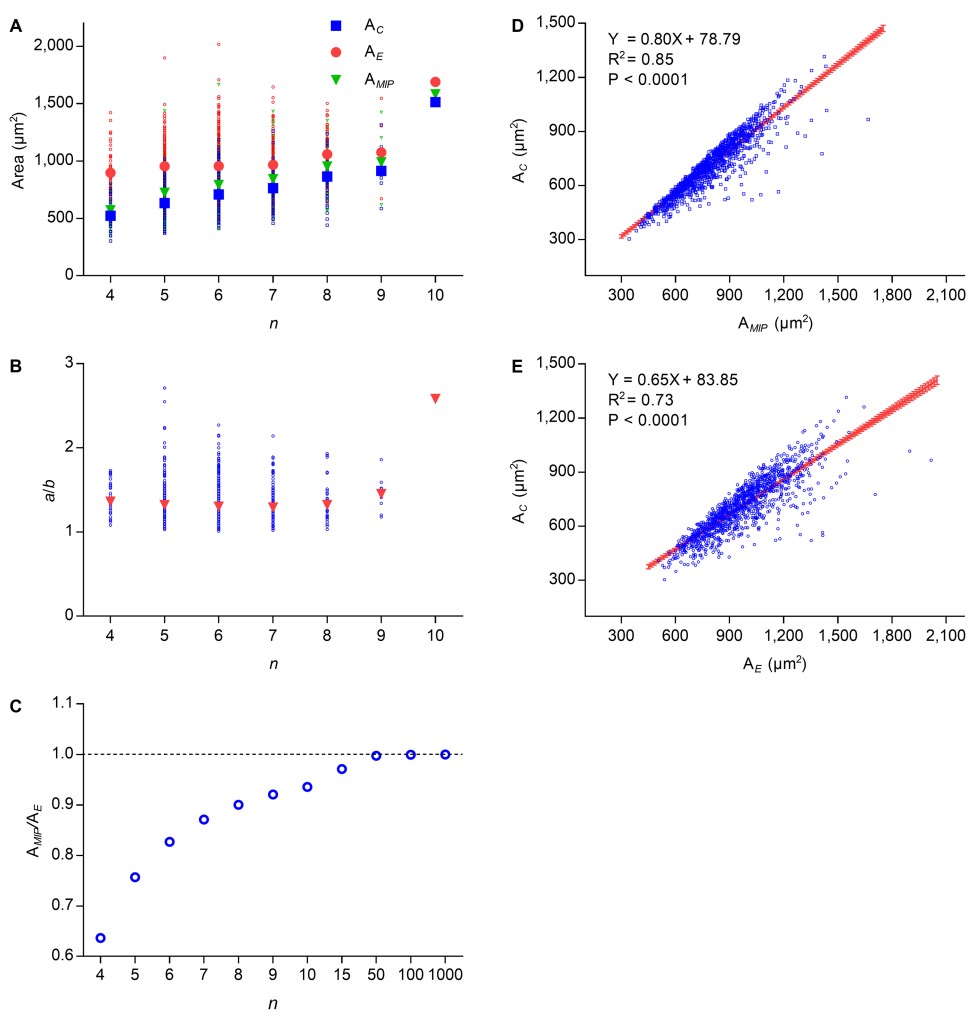

**Figure 2** **Relationships between *n*, A$_C$, A$_{MIP}$, and A$_E$ of *P. haitanensis* cells.** (A) Relationships between the number of cell edges n, area of cell A$_{MIP}$, area of the maximal inscribed polygon A$_{MIP}$, and area of fitted ellipse A$_E$. Big symbols represent the average values of A$_C$, A$_{MIP}$, and A$_E$, whereas small symbols represent the raw data (1375 cells were analyzed). (B) Relationship between n and ratio of a/b. (C) Relationship between n and ratio of A$_{MIP}$/A$_E$. (D) Relationship between A$_C$ and A$_{MIP}$ (1375 cells were analyzed). (E) Relationship between A$_C$ and A$_E$ (1375 cells were analyzed).

transect mother cells at midpoints of unconnected paired edges. Afterward, the direction of the cell edges were changed to concentrate the internal angles ranging from 100° to 140°. Thus, the smallest number of edges per cell was four, and two equal-sized daughter cells were produced.

The ellipse packing is exactly a short-range order, which could influence both local and global cell topology. We used the average axes of the fitted ellipses and average number of edges to calculate the average variation on the internal angles (Table 1, Fig. 3A). Assuming an EMIP with six edges was divided along the minor axis of the ellipse, then ellipse packing should turn all three polygonal cells around the new vertex into EMIPs (Fig. 3B). Thus, two daughter cells would be turned into two five-edged EMIPs with equal sizes, and the

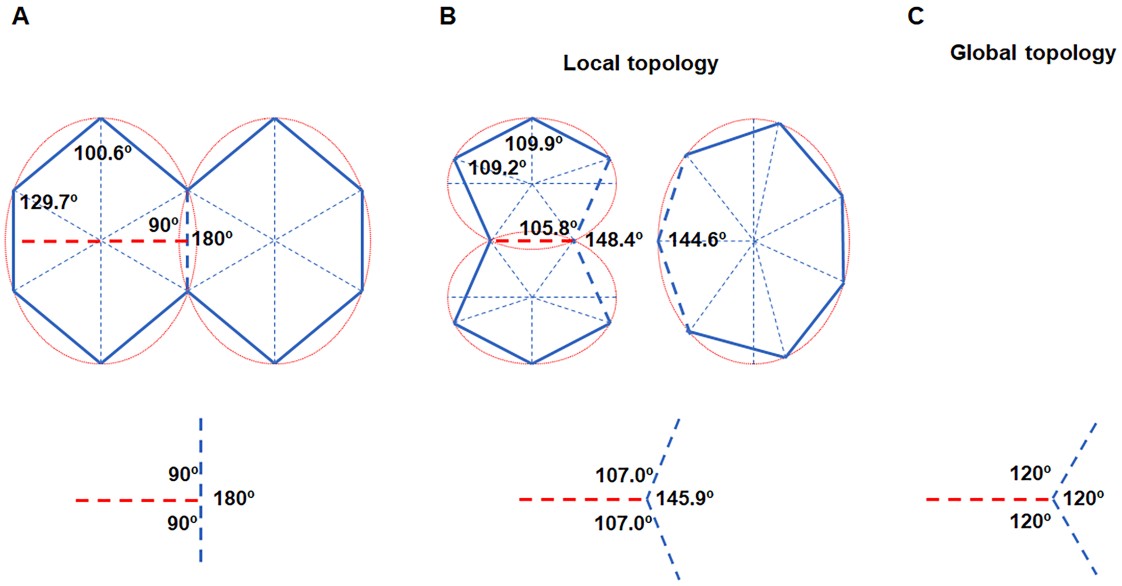

**Figure 3  Cell division obeys ellipse packing.** (A) Red dashed lines represent that division of the maximally inscribed six-gon divided the cell along the minor axis of the ellipse and produced two equal-size daughters. Blue dashed line shows that an edge was separated by a new vertex, which produced three new angles (bottom). (B) Ellipse packing turned the two daughters into maximally inscribed five-gons (top left) by allometric growth of cell edges, whereas the neighboring seven-gon also turned into an EMIP (top right). To minimize the total disruption on the three angles, the turning angle in the neighboring cell should be 34.1° (bottom). (C) The three angles around each vertex tended to be 120°. The ratios of $a/b$ of all of the ellipses were set to an average value of 1.3 (Table 1).

neighboring cell of the daughters would be turned into a seven-edged EMIP. The sum of the three angles around a vertex is 360°. Assuming the total disruptions on the three angles is kept to a minimum, on the basis of the least square method, the newly formed internal angle in the neighboring cell would be decreased from 180° to 145.9°. This would explain the observation that the turning angle was $40 \pm 6°$ (138 angles were examined) in the previous study by *Xu et al. (2017)*. Meanwhile, those angles inherited from the mother cells also had to be adjusted to obey ellipse packing. Obviously, all of these changes on angles must be achieved by allometric growth of the cell edges. The long-axis division could help the cells retain their shapes closest to EMIPs. Finally, from a global perspective, the combined effect of ellipse packing and other short-range order (vertex coordination number is equal to three) turned all three angles around each vertex to 120° (Fig. 3C). Overall, for biological 2D structures, ellipse packing placed restrictions on the direction of cell division and the turning angles of the cell edges.

## Aboav-Weaire's law

If $m$ represents the average number of edges of cells surrounding an $n$-edged cell, then the relation between $m$ and $n$ of Type I 2D structures is $m = n = 6$. As for Type II and III 2D structures, we used Aboav-Weaire's law to describe the relation between $m$ and $n$:

$$m = (6 - \beta) + \frac{6\beta + \mu_2}{n}, \tag{8}$$

where six is the average number of cell edges of 2D structures, $\beta$ is a constant, and $\mu_2$ is related to the second moment of the edges of the $n$-edged cell (*Weaire & Rivier, 1984*). The present study and a previous study by *Xu et al. (2017)* showed that all cells tended to form regular polygons, which indicated that the internal angles of a cell tended to be close to each other. According to Lewis's law, the cell area of Type II and III 2D structures increase with $n$. The average internal angle of an $n$-edged cell is $\pi - \frac{2\pi}{n}$, which also increases with $n$. The sum of three angles around each vertex is $2\pi$, which suggests that the average neighboring angle of the $n$-edged cell is decreasing with an increase of $n$. Consequently, the $m$, the average area, and the average internal angle of neighboring cells tend to decrease with an increase of $n$. Thus, Aboav-Weaire's law describes the representative level of a data set with $2n$ neighboring angles in the total data set with $mn$ internal angles of the neighboring cells. In addition, the mean value of $m$ should also be equal to six.

On the basis of experimental studies, $\beta \approx 1.2$ was found to be conserved for several natural physical and biological structures(*Aboav, 1983*; *Aboav, 1980*; *Mombach, Vasconcellos & De Almeida, 1990*; *Mombach, De Almeida & Iglesias, 1993*). This number was very close to the average ratio of $a/b$ of cells of several kinds of 2D structures (Tables 1–2, Fig. 2B) and of the oval-shaped exoskeletons (faces) of *E. huxleyi* cells (*Xu, Hutchins & Gao, 2018*). In previous studies, $\mu_2$ has been assumed to be small (*Edwards & Pithia, 1994*; *Lambert & Weaire, 1981*). Regular hexagons could monohedrally tessellate a plane (*Grünbaum & Shephard, 1987*). This kind of tessellation also featured with ellipse packing and every vertex had a coordination number equal to three. This indicated that when $n = \langle n \rangle = 6$, $\mu_2 = 0$, where $\langle n \rangle$ is the average number of cell edges. This study assumed

$$\mu_2 = \frac{6-n}{12}. \tag{9}$$

Thus, using Eq. (8), we have

$$m = \left(6 - \frac{a}{b}\right) + \frac{\frac{6a}{b} + \frac{6-n}{12}}{n}, \tag{10}$$

where $a$ and $b$ are the semi-major axis and semi-minor axis of fitted ellipse of an $n$-edged cell, respectively. Then, Eq. (10) can be rewritten as follows:

$$m = 6 + \frac{6-n}{n} \times \left(\frac{a}{b} + \frac{1}{12}\right). \tag{11}$$

This equation could explain the relation between $m$ and $n$ of all three types of 2D structures. The calculated $m$ of cells of Types II and III 2D structures were very close to the real values by enumeration (Fig. 4A). The average difference between calculated $m$ and real $m$ was $-0.13 \pm 0.31$ (371 cells were examined). Because $\mu_2$ is very small, Aboav-Weaire's law could be approximately expressed as follows:

$$m \approx 6 + \frac{6-n}{n} \times \frac{a}{b}. \tag{12}$$

The calculated $m$ using Eqs. (11) and (12) showed only minor differences (Fig. 4A, Data S3). In addition, this study found an empirical relation for Type III 2D structures

$$\frac{1}{12} = 1 - \frac{A_C}{A_{MIP}}, \tag{13}$$

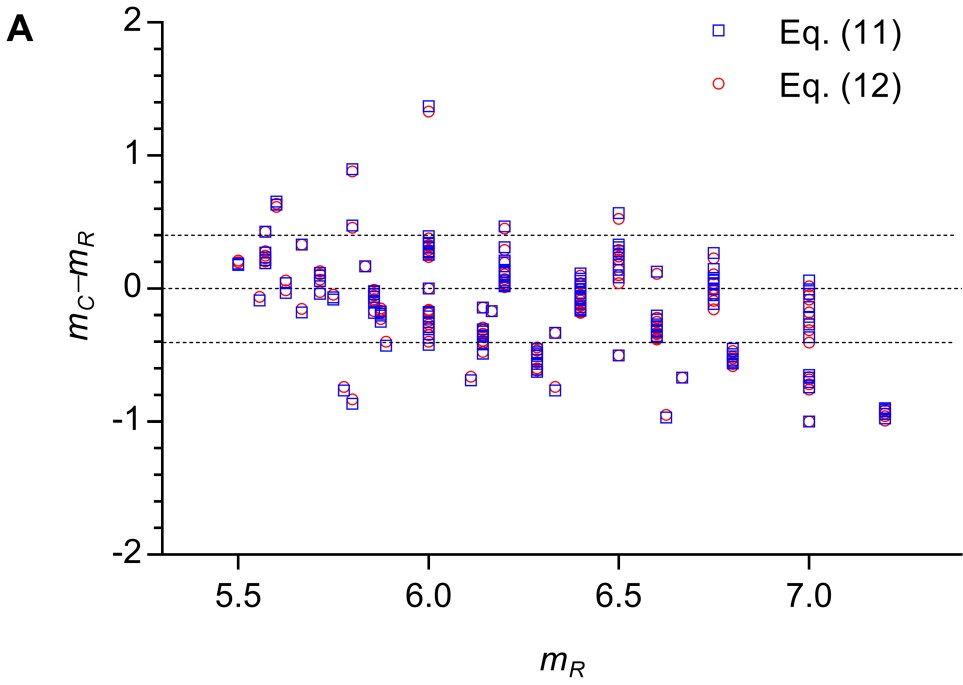

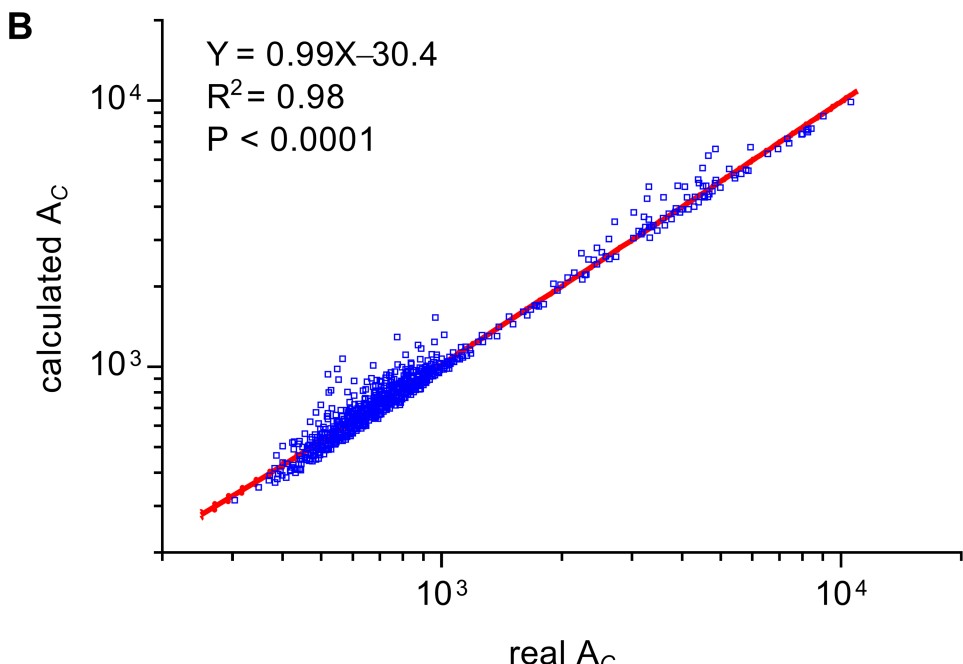

**Figure 4** **Examinations of the relations of Lewis's law and Aboav-Weaire's law.** (A) Relationship between the real m ($m_R$) and the calculated m ($m_C$) of an n-edged cell (371 cells were examined). We used Eqs. (11) and (12) to calculate m. (B) Relationship between real and calculated area ($A_C$) of an n-edged polygonal cell of all Type III 2D structures (1475 cells were examined). We used Eq. (14) to calculate $A_C$. The units of the cell area of *P. haitanensis* (1375 cells, Table 1) and the other Type III 2D structures (100 cells, Table 2) were $\mu m^2$ and pixel$^2$, respectively.

which can be expressed as follows

$$A_C = A_{MIP}\left(1 - \frac{1}{12}\right) = 0.5nab\sin\left(\frac{2\pi}{n}\right)\left(1 - \frac{1}{12}\right). \tag{14}$$

The slope of the relationship between calculated $A_C$ and measured $A_C$ of cells of Type III 2D structures was very close to one ($R^2 = 0.98$, $P < 0.0001$, Fig. 4B). The $a/b$ describes the deformation degree from circle to ellipse. Similarly, the present study proposed that, for Type III 2D structures, the number 1/12 describes the deformation degree from EMIP to EIP. Meanwhile, for Type III 2D structures, the Eq. (12) can be rewritten as follows:

$$m = 6 + \frac{6-n}{n} \times \left(\frac{a}{b} + 1 - \frac{A_C}{A_{MIP}}\right). \tag{15}$$

## Variations of 2D topology

We discussed the variations of cell topology of biological 2D structures in the previous sections. The random-seeded Voronoi diagrams are also Type III 2D structures, which were used to simulate the static structure of biological materials (*Honda, 1983*; *Sánchez-Gutiérrez et al., 2016*). This study found that the ellipse packing creates a strong restriction on cellular geometry (e.g., the edge length and internal angle), which indicates that ellipse packing could be used to predict the effects of cell proliferation on cellular geometry. Thus, the combination of ellipse packing and Voronoi diagram may be applied to simulate the topological dynamic behaviors during the growth of biological 2D structures.

Although the average number of cell edges was always six, the distributions of the edge numbers showed big differences between the 2D materials and varied during growth (*Aboav, 1980*; *Aboav, 1985*; *Büchner et al., 2016*; *Reich et al., 2018*; *Xu et al., 2017*). Moreover, we know little about the relationships between the range of edge numbers and other topological parameters. In this study, we used the interval length (L) to describe the differences between the maximal and minimum edge numbers. Then, Type I 2D structures had the smallest L of 0. During the evolvement of soap, L increased from 3 to 11 (*Aboav, 1980*). Similar phenomena also have been reported in physical 2D materials. For example, the point defects in hexagonal networks, one kind of local variations of topology, manifested with an increase of L from 0 to 2 (*Büchner & Heyde, 2017*).

To avoid confusion caused by the effects of observation scales, we discuss only the topological variations of a 2D physical material with constant mass, and these variations would not influence the connection pattern between atoms. Under the restrictions of ellipse packing and coordination number, the average number of cell edges was always six and the number of cells remained constant. We proposed five basic topological variations (Table 3):

V1. Reconstruction, which will not change the global topological parameters of the 2D structure but will create a new 2D structure with completely changed local topology. From a global scale, the area of the 2D structure will not be changed—for example, the destruction and rebuilding of graphene used the same number of carbon atoms.

**Table 3   Five kinds of basic topology variations of 2D physical structures with constant mass.**

|        | Parameters | V1 | V2 | V3 | V4 | V5 |
|--------|------------|----|----|----|----|----|
| Global | Type of 2D structure | × | × | × or ✓ | ✓ | ✓ |
|        | Area of 2D structure | × | ✓ | × | ✓ | ✓ |
|        | Interval length of range of edge number (L) | × | × | ✓ | × | ✓ |
| Local  | Number of cell edges ($n$) | ✓ | × | ✓ | × | ✓ |
|        | Area of cell ($A_C$) | ✓ | ✓ | ✓ | ✓ | ✓ |
|        | $ab$ | ✓ | ✓ | ✓ | × | ✓ |
|        | $a/b$ | ✓ | × or ✓ | ✓ | × | ✓ |
|        | Average number of edges of neighboring cells ($m$) | ✓ | × | ✓ | × | ✓ |

**Notes.**

V1, Reconstruction; V2, Scaling; V3, L-Variation; V4, Transition between Type II and Type III; V5, Transition between Type I and Type III.

Symbol × represents the parameter will not be changed, and ✓ represents the parameter will be changed.

V2. Scaling, which will not influence the type, L, $n$, and $m$, but the area of the 2D structure and individual cells will be changed. The uniform scaling of ellipses has to be achieved by a uniform change in the edge lengths to maintain constant $ab$ and $a/b$.

V3. L-Variation of Type II, which was featured with a varied L of the Type II 2D structure and will not influence only the area of the 2D structure. Because we considered the Type I 2D structure to be a specific case of Type II with L equal to 0, the transition between Type I and Type II 2D structures actually belonged to L-Variation. For 2D amorphous $SiO_2$ film, the numbers of cell edges ranged from four to nine (*Büchner & Heyde, 2017*; *Büchner et al., 2016*), which indicated three intermediate states (L = 4, 3, and 2) occurred during the transition between crystalline (Type I) and amorphous (Type II) $SiO_2$ film.

V4. Transition between Type II and Type III, which will change the area of 2D structure and individual cells. According to the new equation of Lewis's law in this study, the area of 2D structure should change by 1/12 times, as should the volume if the height of 2D layers remained constant.

V5. Transition between Type I and Type III, which will change all parameters. The area of 2D structure changed by 1/12 times.

Given the variations related to Types I and II 2D structures, the topological variations of all involved cells need to be synchronously finished to obey ellipse packing. The V2, V4, and V5 could influence the area of 2D materials, which may be the most noticeable characteristic of these topological variations. The combination of these five basic topological variations would make it more difficult to understand the overall topological behavior, and the complexity of structure (e.g., heterogeneous materials, dimensionality of material) added further difficulties. For example, in the Voronoi diagrams with spiral lattice, the cells were arranged in a pattern of Fibonacci numbers (*Rivier et al., 1984*; *Rivier, Sadoc & Charvolin, 2016*). More work is needed to deeply understand the ellipse packing and its effects on global and local topology of 2D structures.

## 3D structures

We considered every prismatic cell of the biological 2D structures to be a convex polyhedron with an average face number of eight. On the basis of a model study on 3D Voronoi froth

with random seeds, if the coordination number of multi-polyhedral-celled 3D structures is four, then the average face number is $\left(\frac{48}{35}\right)\pi^2 + 2 (\approx 15.54)$ (*Meijering, 1953*; *Weaire & Rivier, 1984*). This number was very close to the average face number of 15.4 in the polyhedral cells of single-celled microalga *E. huxleyi* with a vertex coordination number of three (*Xu, Hutchins & Gao, 2018*). The difference in the average face number indicated that these 3D structures could not simplified as 2D structures. A convex polyhedral cell is a sealed 3D structure that has a positive curvature at every vertex and obeys Euler's law. Euler's law, however, does not set any restriction on six-edged faces (*Grünbaum & Motzkin, 1963*; *Xu, Hutchins & Gao, 2018*). This suggests that a given 3D structure does not necessarily need to be a sealed structure even it obeys Euler's law. The closure of polyhedra could be considered to be a basic level of uniform distribution of curvature. The face topology of polyhedra could be analyzed using software CaGe (*Brinkmann et al., 2010*).

Polygons with more than six edges induce locally negative curvature, and those with less than six edges induce positive curvature (*Cortijo & Vozmediano, 2007*). Thus, the polyhedral cells of *E. huxleyi* contained four-gons, five-gons, and six-gons, which helped maintain full coverage on the spherical surface (*Xu, Hutchins & Gao, 2018*). As for 2D tessellation using different-sized cells, the average edge number of six determined that the top and bottom faces of *P. haitanensis* cells contained four to ten edges (Table 1). Because of geometric limits, Lewis's law and Aboav-Weaire's law remained valid for the face topology of cells of *E. huxleyi* (*Xu, Hutchins & Gao, 2018*). For living biological 3D structures, as with the living biological 2D structures, growth influenced the topology of the polyhedral cells. Thus, on the basis of this study and the previous study by *Xu, Hutchins & Gao (2018)*, we suggested that the faces of the polyhedral cells would be EIPs rather than EMIPs to allow the cell topology to accommodate complicated life activities, which indicated that Lewis's law for Type III 2D structures (Eq. (14)) also may be applied to living biological 3D structures. Aboav-Weaire's law may be generalized to 3D structures with consideration for the distribution of curvature at vertices.

## CONCLUSION

This study found that polygonal cells of natural and artificial 2D structures were inclined to form EMIPs. This phenomenon was named ellipse packing, which could be applied in simulations of the dynamics of cell topology during growth. We derived improved relations of Lewis's law and Aboav-Weaire's law and verified these findings using the semi-axes of fitted ellipses, cell area, and the number of cell edges. The present study suggested that Lewis's law and Aboav-Weaire's law are nonlinear relations, which mainly describe the effect of circle deformation on cell area and the edge number of neighboring cells. Ellipse packing determines the cell topology of 2D structures and growth patterns.

## ACKNOWLEDGEMENTS

I would like to thank Zhixue Chen and Fangyu Guo for their assistance collecting the geometric data of the cells. I would also like to thank my mother Yuntao Yan and my

wife Huimin Cheng for their wholehearted support. Thanks to Accdon for its linguistic assistance during the preparation of this manuscript.

### Funding

This work was supported by the National Key Research and Development Program of China (2018YFD0900702). The funders had no role in study design, data collection and analysis, decision to publish, or preparation of the manuscript.

### Grant Disclosures

The following grant information was disclosed by the author:
National Key Research and Development Program of China: 2018YFD0900702.

### Competing Interests

The authors declare there are no competing interests.

### Author Contributions

- Kai Xu conceived and designed the experiments, performed the experiments, analyzed the data, contributed reagents/materials/analysis tools, prepared figures and/or tables, authored or reviewed drafts of the paper, approved the final draft.

### Data Availability

   All the raw data are available as Supplementary Files.

### Supplemental Information

Supplemental information for this article can be found online at http://dx.doi.org/10.7717/peerj.6933#supplemental-information.

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
