# Peer review of "Ellipse packing in two-dimensional cell tessellation: a theoretical explanation for Lewis’s law and Aboav-Weaire’s law"

_PeerJ, doi:10.7717/peerj.6933_

## Round 0.1 · original submission · Minor Revisions

Only minor revisions are needed, no further review is necessary once the few issues pointed out by the reviewers have been addressed.

Reviewer 1 ·

Basic reporting

The work is clear and well-written. The background is well presented and the structure is good. It is self-contained.

Experimental design

The question posed is interesting. The approach original. The work is of high standard and of interest to a wide range of problems.

Validity of the findings

The findings are robust and interesting.

Reviewer 2 ·

Basic reporting

.

Experimental design

.

Validity of the findings

.

Additional comments

This paper investigate cells structures as ellipse packings and thereby provide insight into a set of observed laws for such structures.
The investigation seems sound and the results reasonable. What remains unclear, however, is in what way these results are useful for understanding cellular structures.
It is mentioned in the text that the results can be used for simulating cell topology during growth. How this should be done remain unclear, however. In general, it needs to be demonstrated,
or, at least, argued for in what way the results are useful in this respect before the paper can be considered for publication.

Reviewer 3 ·

Basic reporting

.

Experimental design

.

Validity of the findings

.

Additional comments

This paper presents a range of statistical data on 2d foams. It adopts a novel method of analysing 2d cellular patterns, involving the fitting of ellipses to individual cell outlines. I was not persuaded of the necessity for this and indeed some of the conclusions, but I do recommend publication. The work is substantial, and well placed in relation to the literature. Moreover it takes data from a range of biological materials. This will make it useful to many people, particularly as the question of the relation of biological cells to analogous physical systems is topical at this time.

---

## Round 0.2 · accepted · Accept

Thank you for sending the revised version. I am pleased to accept your manuscript.

#